# Risk factors for toxoplasmosis in people living with HIV in the Asia-Pacific region

Ki Hyun Lee[1], Awachana Jiamsakul[2], Sasisopin Kiertiburanakul[3], Rohidas Borse[4], Vohith Khol[5], Evy Yunihastuti[6], Iskandar Azwa[7], I. Ketut Agus Somia[8], Romanee Chaiwarith[9], Thach Ngoc Pham[10], Suwimon Khusuwan[11], Cuong Duy Do[12], Nagalingeswaran Kumarasamy[13], Yasmin Gani[14], Rossana Ditangco[15], Oon Tek Ng[16], Sanjay Pujari[17], Man Po Lee[18], Anchalee Avihingsanon[19], Hsin-Pai Chen[20], Fujie Zhang[21], Junko Tanuma[22], Jeremy Ross[23], Jun Yong Choi[1] *

1 Department of Internal Medicine and AIDS Research Institute, Yonsei University College of Medicine, Seoul, South Korea, 2 The Kirby Institute, UNSW Sydney, Kensington, New South Wales, Australia, 3 Faculty of Medicine Ramathibodi Hospital, Mahidol University, Bangkok, Thailand, 4 BJ Government Medical College and Sassoon General Hospital, Pune, India, 5 National Center for HIV/AIDS, Dermatology & STDs, Phnom Penh, Cambodia, 6 Faculty of Medicine Universitas Indonesia - Dr. Cipto Mangunkusumo General Hospital, Jakarta, Indonesia, 7 Infectious Diseases Unit, Department of Medicine, University of Malaya, Kuala Lumpur, Malaysia, 8 Faculty of Medicine, Udayana University - Prof. Dr. I.G.N.G. Ngoerah Hospital, Bali, Indonesia, 9 Division of Infectious Diseases and Tropical Medicine, Department of Medicine, Faculty of Medicine and Research Institute for Health Sciences, Chiang Mai University, Chiang Mai, Thailand, 10 National Hospital for Tropical Diseases, Hanoi, Vietnam, 11 Chiangrai Prachanukroh Hospital, Chiang Rai, Thailand, 12 Bach Mai Hospital, Hanoi, Vietnam, 13 CART CRS, Voluntary Health Services, Chennai, India, 14 Hospital Sungai Buloh, Sungai Buloh, Malaysia, 15 Research Institute for Tropical Medicine, Muntinlupa City, Philippines, 16 National Centre for Infectious Diseases, Tan Tock Seng Hospital, Singapore, Singapore, 17 Institute of Infectious Diseases, Pune, India, 18 Queen Elizabeth Hospital, Yau Ma Tei, Hong Kong SAR, 19 HIV-NAT/ Thai Red Cross AIDS Research Centre and Center of Excellence in Tuberculosis, Faculty of Medicine, Chulalongkorn University, Bangkok, Thailand, 20 Taipei Veterans General Hospital, Taipei, Taiwan, 21 Beijing Ditan Hospital, Capital Medical University, Beijing, China, 22 National Center for Global Health and Medicine, Tokyo, Japan, 23 TREAT Asia, amfAR - The Foundation for AIDS Research, Bangkok, Thailand

* SERAN@yuhs.ac

**Data Availability Statement:** The following restrictions on sharing a study data should be noted: Study data is available on request but is not publicly available without restriction, as it might

## Abstract

### Introduction

*Toxoplasma gondii* can cause symptomatic toxoplasmosis in immunodeficient hosts, including in people living with human immunodeficiency virus (PLWH), mainly because of the reactivation of latent infection. We assessed the prevalence of toxoplasmosis and its associated risk factors in PLWH in the Asia-Pacific region using data from the TREAT Asia Human Immunodeficiency Virus (HIV) Observational Database (TAHOD) of the International Epidemiology Databases to Evaluate AIDS (IeDEA) Asia-Pacific.

### Methods

This study included both retrospective and prospective cases of toxoplasmosis reported between 1997 and 2020. A matched case-control method was employed, where PLWH diagnosed with toxoplasmosis (cases) were each matched to two PLWH without a toxoplasmosis diagnosis (controls) from the same site. Sites without toxoplasmosis were excluded. Risk factors for toxoplasmosis were analyzed using conditional logistic regression.

contain potentially identifying information (e.g. date of birth), and because study data is considered owned by the study sites contributing data. These data sharing restrictions are imposed by the TAHOD study group, including participating site principle investigators. Data requests may be sent to the study data manager, Awachana Jiamsakul (Ajiamsakul@kirby.unsw.edu.au), at the Kirby Institute, UNSW, Sydney, Australia, which serves as the study's regional data center. We appreciate your updating of our Data Availability statement to reflect this information.

**Funding:** The authors received no specific funding for this work.

**Competing interests:** The authors have declared that no competing interests exist.

**Abbreviations:** AIDS, acquired immune deficiency syndrome; ART, antiretroviral therapy; CDC, Centers for Disease Control and Prevention; HBV, hepatitis B virus; HCV, hepatitis C virus; HIV, Human Immunodeficiency Virus; IDU, injection drug use; IQR, Interquartile range; MSM, men who have sex with men; NNRTI, non-nucleoside reverse transcriptase inhibitors; NRTI, nucleoside reverse transcriptase inhibitors; OR, odds ratio; PI, protease inhibitor; PLWH, people living with HIV; TAHOD, TREAT Asia HIV Observational Database; VL, viral load.

## Results

A total of 269/9576 (2.8%) PLWH were diagnosed with toxoplasmosis in 19 TAHOD sites. Of these, 227 (84%) were reported retrospectively and 42 (16%) were prospective diagnoses after cohort enrollment. At the time of toxoplasmosis diagnosis, the median age was 33 years (interquartile range 28–38), and 80% participants were male, 75% were not on antiretroviral therapy (ART). Excluding 63 out of 269 people without CD4 values, 192 (93.2%) had CD4 ≤200 cells/μL and 162 (78.6%) had CD4 ≤100 cells/μL. By employing 538 matched controls, we found that factors associated with toxoplasmosis included abstaining from ART (odds ratio [OR] 3.62, 95% CI 1.81–7.24), in comparison to receiving nucleoside reverse transcriptase inhibitors plus non-nucleoside reverse transcriptase inhibitors, HIV exposure through injection drug use (OR 2.27, 95% CI 1.15–4.47) as opposed to engaging in heterosexual intercourse and testing positive for hepatitis B virus surface antigen (OR 3.19, 95% CI 1.41–7.21). Toxoplasmosis was less likely with increasing CD4 counts (51–100 cells/μL: OR 0.41, 95% CI 0.18–0.96; 101–200 cells/μL: OR 0.14, 95% CI 0.06–0.34; >200 cells/μL: OR 0.02, 95% CI 0.01–0.06), when compared to CD4 ≤50 cells/μL. Moreover, the use of prophylactic cotrimoxazole was not associated with toxoplasmosis.

## Conclusions

Symptomatic toxoplasmosis is rare but still occurs in PLWH in the Asia-Pacific region, especially in the context of delayed diagnosis, causing advanced HIV disease. Immune reconstitution through early diagnosis and ART administration remains a priority in Asian PLWH.

## Introduction

The obligate intracellular protozoan parasite *Toxoplasma gondii* is a prevalent pathogen worldwide [1]. Because significant overt infections are rare, the extent of the actual *Toxoplasma* infection is mainly estimated based on serological test results. Seroprevalence varies depending on the region, environment, cultural habits, and lifestyle, with seropositivity rates exceeding 70% in some areas [2].

*Toxoplasma* infection rarely appears as an overt infection despite its high seropositivity. However, in people living with human immunodeficiency virus (PLWH) with low CD4+ T lymphocytes, toxoplasmic encephalitis is the most common cause of central nervous system infection [3, 4]. In patients with acquired immune deficiency syndrome (AIDS) who have a CD4 count <100 cells/μL, toxoplasmosis reactivated in approximately 30% of the patients who were seropositive and not undergoing antiretroviral therapy (ART) [5, 6]. Therefore, screening for anti-*Toxoplasma* antibodies and prophylactic administration of cotrimoxazole are required for managing human immunodeficiency virus (HIV) infection [7–9]. It can also manifest as pneumonitis, retinochoroiditis, or disseminated infections [8, 10]. Seropositivity in PLWH should not be disregarded, as these infections appear mainly as a result of the reactivation of latent infections.

The risk factors for *Toxoplasma* seropositivity, including consumption of raw or undercooked meat, frequent contact with soil, older age, and lower CD4+ lymphocyte counts, have been clarified to some extent [1, 11, 12]. However, some limitations exist in applying these to a special group, such as PLWH, and most studies to date have focused on the risk factors of the

seropositive rate. In addition, in recent years, ART and the prevention of opportunistic infections have been increasing in PLWH; however, studies investigating the corresponding changes in actual *Toxoplasma* infections are insufficient [13].

To this end, this study aimed to investigate the reported toxoplasmosis diagnoses and their risk factors in the Asia-Pacific region using data from the TREAT Asia HIV Observational Database (TAHOD) of the International Epidemiology Databases to Evaluate AIDS (IeDEA) Asia-Pacific.

## Materials and methods

### Data sources and study population

We analyzed data through July 2022 from TAHOD, a prospective observational cohort study of HIV-positive adults enrolled from 21 clinical sites, which contributes to the IeDEA Asia-Pacific [14]. The TAHOD database and methods have been described previously [15]. Study population was selected from patients enrolled in the TAHOD from 2003 to 2021, according to the latest annual data summary. Due to the observational nature of the cohort, laboratory tests were not performed on a predefined basis but depended on the local practices of the site. PLWH from TAHOD sites with reported cases of toxoplasmosis were included (sites without any reported cases were excluded). We included cases of toxoplasmosis matched to two controls without toxoplasmosis from the same site. Ethics approvals for the study were obtained from the coordinating center (TREAT Asia ethics/amfAR, Bangkok, Thailand), the data management and analysis center (University of New South Wales Human Research Ethics Committee, Sydney, Australia). As the pure observational nature of the study, TAHOD waived the written informed consent. And all TAHOD data transfers are anonymized before submission to the Kirby Institute.

### Definition of toxoplasmosis

We defined meaningful toxoplasma infection as definitive and presumptive toxoplasmosis, and definitive toxoplasmosis means patients who confirmed toxoplasmosis by microscopy (histology or cytology). Presumptive toxoplasmosis means patients whom suspicious toxoplasmosis of brain satisfying all three of the following. (i) Recent onset of a focal neurological abnormality consistent with intracranial disease or a reduced level of consciousness; (ii) Evidence by brain imaging (computed tomography or nuclear magnetic resonance) of a lesion having a mass effect or the radiographical appearance of which is enhanced by injection of contrast medium or consistent symptomatic presentation; (iii) Serum antibody to toxoplasmosis or successful response to therapy for toxoplasmosis.

### Matching

A prospective toxoplasmosis case was defined as a reported diagnosis of toxoplasmosis (collected as Centers for Disease Control and Prevention [CDC]-category C in our cohort) after TAHOD enrollment. A retrospective case was defined as a reported diagnosis that occurred prior to entry into the TAHOD. To account for differences in site and time period, prospective cases were matched to prospective controls, while retrospective cases were matched to retrospective controls from the same site and time period. Prospective controls were those without evidence of toxoplasmosis with prospective follow-up data available during the same period (+/- 3 months) as the date of toxoplasmosis diagnosis in the prospective case. Retrospective controls were those without evidence of toxoplasmosis who had retrospective (prior to cohort enrollment) data available within the same timeframe as the retrospective control diagnosis.

The controls were randomly selected and were required to have at least a CD4 or viral load (VL) available within the matched timeframe as they were considered key covariates in the study. Controls without both viral load (VL) and CD4 available were not selected.

## Statistical analysis

Factors associated with toxoplasmosis were analyzed using conditional logistic regression to account for matching. The regression models were fitted using backward stepwise procedures. Factors that were significant in univariate analyses ($p<0.10$) were included in the multivariate analyses. Factors with $p<0.05$ in the final multivariate model were considered statistically significant. Data management and statistical analyses were performed using SAS software (version 9.4; SAS Institute Inc., Cary, NC, USA) and Stata software version 16.1 (Stata Corp., College Station, TX, USA).

## Results

### Demographics and clinical characteristics of the study population

Although our data cannot represent the incidence because of the study design, the years in which the subjects diagnosed with toxoplasma are listed in S1 Fig. A total of 269/9576 (2.8%) PLWH were diagnosed with toxoplasmosis from a subset of 19 TAHOD sites from Cambodia, China, Hong Kong SAR, India, Indonesia, Japan, South Korea, Malaysia, the Philippines, Singapore Taiwan, Thailand, and Vietnam, and were matched to 538 controls. Of the 269 cases, 227 (84%) were reported retrospectively and 42 (16%) were prospective diagnoses after cohort enrollment. Baseline characteristics and demographics are presented in Table 1. At the time of toxoplasmosis diagnosis, the median age was 33 years (interquartile range [IQR] 28–38), 80% participants were male, 75% were not on ART. Excluding 63 out of 269 people without CD4 values, 192 (93.2%) had CD4 $\leq$200 cells/µL and 162 (78.6%) had CD4 $\leq$100 cells/µL. A total of 55 patients (20.4%) who received prophylactic cotrimoxazole, defined as cotrimoxazole use within 3 months prior to the diagnosis of toxoplasmosis, were within the same matched time frame for controls.

### Risk factors for toxoplasmosis

Table 2 shows that factors associated with toxoplasmosis included abstaining from ART (odds ratio [OR] 3.62, 95% confidence interval [CI] 1.81–7.24) in comparison to receiving nucleoside/nucleotide reverse transcriptase (NRTI) plus non-nucleoside reverse transcriptase inhibitors (NNRTIs), acquiring HIV through injection drug use (IDU) (OR 2.27, 95% CI 1.15–4,47) as opposed to engaging in heterosexual intercourse, and testing positive for hepatitis B virus (HBV) surface antigen (OR 3.19, 95% CI 1.41–7.21). Toxoplasmosis was less likely with in patients with higher CD4 counts (51–100 cells/µL: OR 0.41, 95% CI 0.18–0.96; 101–200 cells/µL: OR 0.14, 95% CI 0.06–0.34; >200 cells/µL: OR 0.02, 95% CI 0.01–0.06) compared to in patients with CD4 $\leq$50 cells/µL. Meanwhile, the use of prophylactic cotrimoxazole was not associated with toxoplasmosis.

## Discussion

Toxoplasmosis is one of the most common opportunistic infections, affecting more than one-third of the global population. The extent of infection varies according to region, age distribution, dietary habits, and lifestyle. Although the seroprevalence rate in the general population is low in the Asia-Pacific region, latent infection in PLWH is high in some regions of East Asia [2, 11, 16]. In addition, in PLWH, these latent infections are mainly manifests as *Toxoplasma*

**Table 1. Demographics and clinical characteristics of the study population.**

| | Number of patients (%) | Number of controls without toxoplasmosis (%) | Number of cases with toxoplasmosis (%) |
|---|---|---|---|
| | N = 807 | N = 538 | N = 269 |
| **Age (years)** | Median = 33, IQR (28–40) | Median = 33, IQR (28–40) | Median = 33, IQR (28–38) |
| ≤30 | 300 (37.2) | 205 (38.1) | 95 (35.3) |
| 31–40 | 330 (40.9) | 213 (39.6) | 117 (43.5) |
| 41–50 | 124 (15.4) | 84 (15.6) | 40 (14.9) |
| >50 | 53 (6.6) | 36 (6.7) | 17 (6.3) |
| **Sex** | | | |
| Male | 573 (71.0) | 359 (66.7) | 214 (79.6) |
| Female | 234 (29.0) | 179 (33.3) | 55 (20.4) |
| **Mode of HIV exposure** | | | |
| Heterosexual | 546 (67.7) | 378 (70.3) | 168 (62.5) |
| MSM | 66 (8.2) | 47 (8.7) | 19 (7.1) |
| Injecting drug use | 153 (19.0) | 82 (15.2) | 71 (26.4) |
| Other/Unknown | 42 (5.2) | 31 (5.8) | 11 (4.1) |
| **ART** | | | |
| NRTI+NNRTI | 204 (25.3) | 153 (28.4) | 51 (19.0) |
| NRTI+PI | 27 (3.3) | 18 (3.3) | 9 (3.3) |
| Other | 24 (3.0) | 16 (3.0) | 8 (3.0) |
| None | 552 (68.4) | 351 (65.2) | 201 (74.7) |
| **CD4 (cells/µL)** | Median = 113, IQR (35–301.5) | Median = 268, IQR (132–483) | Median = 40.5, IQR (20–83) |
| ≤50 | 141 (17.5) | 16 (3.0) | 125 (46.5) |
| 51–100 | 56 (6.9) | 19 (3.5) | 37 (13.8) |
| 101–200 | 74 (9.2) | 44 (8.2) | 30 (11.2) |
| >200 | 137 (17.0) | 123 (22.9) | 14 (5.2) |
| Not tested | 399 (49.4) | 336 (62.5) | 63 (23.4) |
| **VL (copies/mL)** | Median = 6660, IQR (49–170000) | Median = 52, IQR (49–15065) | Median = 100000, IQR (6864.5–260000) |
| <1000 | 76 (9.4) | 60 (11.2) | 16 (5.9) |
| ≥1000 | 91 (11.3) | 27 (5.0) | 64 (23.8) |
| Not tested | 640 (79.3) | 451 (83.8) | 189 (70.3) |
| **HBV co-infection** | | | |
| Negative | 577 (71.5) | 390 (72.5) | 187 (69.5) |
| Positive | 64 (7.9) | 35 (6.5) | 29 (10.8) |
| Not tested | 166 (20.6) | 113 (21.0) | 53 (19.7) |
| **HCV co-infection** | | | |
| Negative | 440 (54.5) | 300 (55.8) | 140 (52.0) |
| Positive | 160 (19.8) | 91 (16.9) | 69 (25.7) |
| Not tested | 207 (25.7) | 147 (27.3) | 60 (22.3) |
| **Prior AIDS** | | | |
| No | 434 (53.8) | 434 (80.7) | 0 (0.0) |
| Yes | 373 (46.2) | 104 (19.3) | 269 (100.0) |
| **Prophylactic cotrimoxazole use** | | | |
| No | 664 (82.3) | 450 (83.6) | 214 (79.6) |
| Yes | 143 (17.7) | 88 (16.4) | 55 (20.4) |

Data are presented as the median (interquartile range [IQR]) or number (%) unless otherwise indicated. OR, odds ratio; HIV, human immunodeficiency virus; MSM, men who have sex with men; ART, antiretroviral therapy; NRTI, nucleoside reverse transcriptase inhibitor; NNRTI, non-nucleoside reverse transcriptase inhibitor; PI, protease inhibitor; VL, viral load; HBV, hepatitis B virus; HCV, hepatitis C virus; AIDS; acquired immune deficiency syndrome.

**Table 2. Risk factors associated with toxoplasmosis.**

| | Number of subjects | | Univariate analysis | | | Multivariate analysis | | |
|---|---|---|---|---|---|---|---|---|
| | Control group | Case group | OR | 95% CI | p | OR | 95% CI | p |
| **Total** | 538 | 269 | | | | | | |
| **Age (years)** | | | | | 0.839 | | | |
| ≤30 | 205 | 95 | 1 | | | | | |
| 31–40 | 213 | 117 | 1.18 | (0.85, 1.65) | 0.320 | | | |
| 41–50 | 84 | 40 | 1.03 | (0.65, 1.63) | 0.899 | | | |
| >50 | 36 | 17 | 1.02 | (0.54, 1.92) | 0.950 | | | |
| **Sex** | | | | | | | | |
| Male | 359 | 214 | 1 | | | | | |
| Female | 179 | 55 | 0.49 | (0.34, 0.71) | <0.001 | | | |
| **Mode of HIV exposure** | | | | | <0.001 | | | 0.045 |
| Heterosexual | 378 | 168 | 1 | | | 1 | | |
| MSM | 47 | 19 | 0.87 | (0.48, 1.56) | 0.636 | 1.88 | (0.79, 4.47) | 0.152 |
| Injecting drug use | 82 | 71 | 2.69 | (1.69, 4.28) | <0.001 | 2.27 | (1.15, 4.47) | 0.018 |
| Other/Unknown | 31 | 11 | 0.80 | (0.39, 1.64) | 0.535 | 0.72 | (0.23, 2.29) | 0.580 |
| **ART** | | | | | 0.007 | | | 0.003 |
| NRTI+NNRTI | 153 | 51 | 1 | | | 1 | | |
| NRTI+PI | 18 | 9 | 1.76 | (0.72, 4.31) | 0.213 | 1.96 | (0.49, 7.79) | 0.337 |
| Other | 16 | 8 | 1.97 | (0.77, 5.05) | 0.156 | 1.83 | (0.45, 7.49) | 0.401 |
| None | 351 | 201 | 2.24 | (1.43, 3.53) | <0.001 | 3.62 | (1.81, 7.24) | <0.001 |
| **CD4 (cells/μL)** | | | | | <0.001 | | | <0.001 |
| ≤50 | 16 | 125 | 1 | | | 1 | | |
| 51–100 | 19 | 37 | 0.35 | (0.16, 0.77) | 0.010 | 0.41 | (0.18, 0.96) | 0.039 |
| 101–200 | 44 | 30 | 0.15 | (0.07, 0.32) | <0.001 | 0.14 | (0.06, 0.34) | <0.001 |
| >200 | 123 | 14 | 0.02 | (0.01, 0.05) | <0.001 | 0.02 | (0.01, 0.06) | <0.001 |
| Not tested | 336 | 63 | | | | | | |
| **VL (copies/mL)** | | | | | | | | |
| <1000 | 60 | 16 | 1 | | | | | |
| ≥1000 | 27 | 64 | 9.83 | (4.4, 21.96) | 0.000 | | | |
| Not tested | 451 | 189 | | | | | | |
| **HBV co-infection** | | | | | | | | |
| Negative | 390 | 187 | 1 | | | 1 | | |
| Positive | 35 | 29 | 1.71 | (1.02, 2.86) | 0.042 | 3.19 | (1.41, 7.21) | 0.005 |
| Not tested | 113 | 53 | | | | | | |
| **HCV co-infection** | | | | | | | | |
| Negative | 300 | 140 | 1 | | | | | |
| Positive | 91 | 69 | 1.90 | (1.22, 2.94) | 0.004 | | | |
| Not tested | 147 | 60 | | | | | | |
| **Prior AIDS** | | | | | | | | |
| No | 434 | 0 | N/A | | | | | |
| Yes | 104 | 269 | | | | | | |
| **Prophylactic cotrimoxazole use** | | | | | | | | |
| No | 450 | 214 | 1 | | | | | |
| Yes | 88 | 55 | 1.35 | (0.91, 1.99) | 0.136 | | | |

Data are presented as the median (IQR) or number (%) unless otherwise indicated. OR; odds ratio, HIV; Human immunodeficiency virus, MSM; men who have sex with men, ART; antiretroviral therapy, NRTIs; nucleoside reverse transcriptase inhibitors, NNRTIs; non-nucleoside reverse transcriptase inhibitors, PIs; protease inhibitors, VL; viral load, HBV; hepatitis B virus, HCV; hepatitis C virus, AIDS; acquired immune deficiency syndrome.

encephalopathy, which can lead to mortality and severe sequelae. Low CD4+ lymphocytes are a known risk factor, and active surveillance and evaluation are required in PLWH.

In the present study, we assessed clinically significant toxoplasmosis, including confirmed toxoplasmosis or cerebral toxoplasmosis, which are the most common manifestations of *Toxoplasma* infection in PLWH. *Toxoplasma* seroprevalence differs with differing local environment; therefore, we analyzed the risk factors using site-based matched case-control studies. Consistent with the results of several previous studies, our study showed that the risk of toxoplasmosis increased in patients who did not receive ART compared to those who received NRTI plus NNRTI. Furthermore, CD4 value was lower in the toxoplasmosis group compared to the control. Excluding cases without CD4 test results, the majority had advanced HIV, with 93.2% having CD4 ≤200 cells/μL. Assuming the same seroprevalence as site matching was performed, low CD4 count is a prominent risk factor for overt toxoplasma infection. Therefore, immune reconstitution through early diagnosis and use of ART remains an important task. Also, compared to the heterosexual mode of HIV exposure and HBV co-infection, IDU increased the risk of toxoplasmosis. The precise etiology behind the association of HBV co-infection with toxoplasmosis in our study remains unclear; however, this association may be attributed to socioeconomic status. Socioeconomic status is also associated with the prevalence of HIV and toxoplasmosis, and several studies have strongly suggested the impact of social and economic factors on the prevalence of HBV [17–20]. Although our study showed that HBV co-infection increases the risk of *Toxoplasma* infection, additional research is needed to clarify the exact causal relationship. In the general population, the prevalence of *Toxoplasma* is high in cases of substance abuse or IDU, suggesting that PLWH may also be affected [21, 22]. In addition, HIV-related cerebral toxoplasmosis occurs frequently in PLWH in low-to middle-income countries, which could be attributed to the fact that IDU and HBV co-infection are common in regions with poor economic conditions [3]. Our study did not find an association between age and increased risk of toxoplasmosis, unlike other study [23]. Perhaps because the previous study was a comparative study based on seroprevalence, and because seroconversion occurs due to exposure to various environments as one ages, so it may be difficult to correlate it as an actual meaningful infection. Also, prophylactic cotrimoxazole administration did not contribute to lowering the risk of toxoplasmosis. When we first analyzed cotrimoxazole administration before and after the diagnosis of toxoplasmosis, cotrimoxazole administration was significantly high in the case group. But when limited to prophylactic use, there was no difference from the control group. This suggests that there might be a problem with medication compliance, but our study cannot prove it.

This study has several other limitations. Both retrospective and prospective studies were included to maximize the number of cases. However, we could not determine the true prevalence of toxoplasmosis in our cohort. As most of our cases were diagnosed before cohort entry, with historical data reporting being voluntary, owing to the prospective nature of the cohort, we found it impossible to capture all toxoplasmosis diagnoses that occurred in TAHOD. Therefore, we were unable to determine the annual proportion of toxoplasmosis diagnoses over time. Because toxoplasmosis cases were identified through CDC-C category reporting, we were unable to assess further details of the diagnostic pathology.

## Conclusions

Our findings indicate that symptomatic toxoplasmosis is rare but still occurs in PLWH in the Asia-Pacific region, especially in the context of delayed diagnosis, causing advanced HIV disease. Immune reconstitution through early diagnosis and ART administration remains a priority in Asian PLWH.

## Supporting information

**S1 Fig. Year of toxoplasmosis diagnosis.**
(DOCX)

## Acknowledgments

The TREAT Asia HIV Observational Database is an initiative of TREAT Asia, a program of amfAR, The Foundation for AIDS Research, with support from the U.S. National Institutes of Health's National Institute of Allergy and Infectious Diseases, *Eunice Kennedy Shriver* National Institute of Child Health and Human Development, National Cancer Institute, the National Institute of Mental Health, National Institute on Drug Abuse, the National Heart, Lung, and Blood Institute, National Institute on Alcohol Abuse and Alcoholism, National Institute of Diabetes and Digestive and Kidney Diseases, and Fogarty International Center, as part of the International Epidemiology Databases to Evaluate AIDS (IeDEA; U01AI069907). The Kirby Institute is funded by the Australian Government Department of Health and Aging and is affiliated with the Faculty of Medicine, UNSW, Sydney. The content of this publication is the sole responsibility of the authors and does not necessarily represent the official views of any of the governments or institutions mentioned above.

## TAHOD study members

V Khol*, V Ouk, C Pov: National Center for HIV/AIDS, Dermatology & STDs, Phnom Penh, Cambodia; FJ Zhang*, HX Zhao, N Han: Beijing Ditan Hospital, Capital Medical University, Beijing, China; MP Lee*, PCK Li, TS Kwong, TH Li: Queen Elizabeth Hospital, Hong Kong SAR; N Kumarasamy*, C Ezhilarasi: Chennai Antiviral Research and Treatment Clinical Research Site (CART CRS), VHS-Infectious Diseases Medical Centre, VHS, Chennai, India; S Pujari*, K Joshi, S Gaikwad, A Chitalikar: Institute of Infectious Diseases, Pune, India; RT Borse*, V Mave, I Marbaniang, S Nimkar: BJ Government Medical College and Sassoon General Hospital, Pune, India; IKA Somia*, TP Merati, AAS Sawitri, F Yuliana: Faculty of Medicine Udayana University—Prof. Dr. I.G.N.G. Ngoerah Hospital, Bali, Indonesia; E Yunihastuti*, A Widhani, S Maria, TH Karjadi: Faculty of Medicine Universitas Indonesia—Dr. Cipto Mangunkusumo General Hospital, Jakarta, Indonesia; J Tanuma*, S Oka, H Uemura, Y Koizumi: National Center for Global Health and Medicine, Tokyo, Japan; JY Choi*, Na S, JM Kim: Division of Infectious Diseases, Department of Internal Medicine, Yonsei University College of Medicine, Seoul, South Korea; YM Gani*, NB Rudi: Hospital Sungai Buloh, Sungai Buloh, Malaysia; I Azwa*, A Kamarulzaman, SF Syed Omar, S Ponnampalavanar: University Malaya Medical Centre, Kuala Lumpur, Malaysia; R Ditangco*, MK Pasayan, ML Mationg: Research Institute for Tropical Medicine, Muntinlupa City, Philippines; HP Chen*, YJ Chan, PF Wu: Taipei Veterans General Hospital, Taipei, Taiwan; OT Ng*, PL Lim, LS Lee, T Yap: Tan Tock Seng Hospital, National Centre for Infectious Diseases, Singapore (note: OT Ng is also supported by the Singapore Ministry of Health's (MOH) National Medical Research Council (NMRC) Clinician Scientist Award (MOH-000276). Any opinions, findings, conclusions, or recommendations expressed in this material are those of the author(s) and do not reflect the views of MOH/NMRC); A Avihingsanon*, S Gatechompol, P Phanuphak, C Phadungphon: HIV-NAT/Thai Red Cross AIDS Research Centre, Bangkok, Thailand; S Kiertiburanakul*, A Phuphuakrat, L Chumla, N Sanmeema: Faculty of Medicine Ramathibodi Hospital, Mahidol University, Bangkok, Thailand; R Chaiwarith*, T Sirisanthana, J Praparattanapan, K Nuket: Faculty of Medicine and Research Institute for Health Sciences, Chiang Mai University, Chiang Mai, Thailand; S Khusuwan*, P Kambua, S Pongprapass, J Limlertchareonwanit: Chiangrai

Prachanukroh Hospital, Chiang Rai, Thailand; TN Pham*, KV Nguyen, DTH Nguyen, DT Nguyen: National Hospital for Tropical Diseases, Hanoi, Vietnam; CD Do*, AV Ngo, LT Nguyen: Bach Mai Hospital, Hanoi, Vietnam; AH Sohn*, JL Ross*, B Petersen: TREAT Asia, amfAR—The Foundation for AIDS Research, Bangkok, Thailand; MG Law*, A Jiamsakul*, D Rupasinghe: The Kirby Institute, UNSW Sydney, NSW, Australia.

\* TAHOD Steering Committee Members

## Author Contributions

**Conceptualization:** Nagalingeswaran Kumarasamy, Jun Yong Choi.

**Data curation:** Awachana Jiamsakul, Sasisopin Kiertiburanakul.

**Writing – original draft:** Ki Hyun Lee.

**Writing – review & editing:** Ki Hyun Lee, Awachana Jiamsakul, Sasisopin Kiertiburanakul, Rohidas Borse, Vohith Khol, Evy Yunihastuti, Iskandar Azwa, I. Ketut Agus Somia, Romanee Chaiwarith, Thach Ngoc Pham, Suwimon Khusuwan, Cuong Duy Do, Yasmin Gani, Rossana Ditangco, Oon Tek Ng, Sanjay Pujari, Man Po Lee, Anchalee Avihingsanon, Hsin-Pai Chen, Fujie Zhang, Junko Tanuma, Jeremy Ross, Jun Yong Choi.

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
