## [Decision Letter · Decision Letter 0]

17 Nov 2023

PONE-D-23-29849Risk factors for toxoplasmosis in people living with HIV in the Asia-Pacific regionPLOS ONE

Dear Dr. Choi,

Thank you for submitting your manuscript to PLOS ONE. After careful consideration, we feel that it has merit but does not fully meet PLOS ONE’s publication criteria as it currently stands. Therefore, we invite you to submit a revised version of the manuscript that addresses the points raised during the review process.

**ACADEMIC EDITOR:**

We look forward to receiving your revised manuscript.

Kind regards,

Masoud Foroutan, Ph.D; Assistant Professor

Academic Editor

PLOS ONE

“The TREAT Asia HIV Observational Database is an initiative of TREAT Asia, a program of amfAR, The Foundation for AIDS Research, with support from the U.S. National Institutes of Health’s National Institute of Allergy and Infectious Diseases, Eunice Kennedy Shriver National Institute of Child Health and Human Development, National Cancer Institute, the National Institute of Mental Health, National Institute on Drug Abuse, the National Heart, Lung, and Blood Institute, National Institute on Alcohol Abuse and Alcoholism, National Institute of Diabetes and Digestive and Kidney Diseases, and Fogarty International Center, as part of the International Epidemiology Databases to Evaluate AIDS (IeDEA; U01AI069907). The Kirby Institute is funded by the Australian Government Department of Health and Aging and is affiliated with the Faculty of Medicine, UNSW, Sydney. The content of this publication is the sole responsibility of the authors and does not necessarily represent the official views of any of the governments or institutions mentioned above.”

“The authors received no specific funding for this work.”

Reviewers' comments:

Reviewer's Responses to Questions

**Comments to the Author**

1. Is the manuscript technically sound, and do the data support the conclusions?

Reviewer #1: No

Reviewer #2: Partly

2. Has the statistical analysis been performed appropriately and rigorously? 

Reviewer #1: No

Reviewer #2: No

3. Have the authors made all data underlying the findings in their manuscript fully available?

Reviewer #1: Yes

Reviewer #2: No

4. Is the manuscript presented in an intelligible fashion and written in standard English?

Reviewer #1: Yes

Reviewer #2: Yes

5. Review Comments to the Author

Reviewer #1: 1. Because patient geographical distribution and infection period difference are also critical for toxoplasmosis infection, the manuscript needs to analyze these data.

2. Because many patients have not taken ART and lack CD4 & viral load data, these data deficiencies could interfere with the risk factors assessment.

3. The IVDU etiology and HBV infection should be considered as counfounding factors and these factors are highly associated to socioeconomic status which could be the risk factor of toxoplasmosis infection.

4. The disease type of toxoplasmosis should be described in this manuscript.

Reviewer #2: In this paper, Hyun Lee et all present a retrospective/prospective case control study to assess risk factors for toxoplasmosis in PLWH in the ASIA Pacific region. With the current use of ART worldwide, opportunistic infections associated to AIDS are infrequent and deserve less attention. Thus, the effort of the authors in conducting this study should be recognized.

However, the study has some important limitations.

-Despite in the Methods section they comment that “controls without VL or CD4 unavailable within the matched timeframe were not selected because they were considered key covariates”, 62.5% and 83.8% of controls were not tested for CD4 or VL respectively (compared with 23.4% and 70.3% in cases). Could the authors clarify this relevant aspect of their study?

-In the abstract they say that 71% of toxo pts had CD4 < 200 , that means that 29% had >200, what is surprising as almost all PLWH with toxo have CD4 < 200 and mostly <100. But in fact, as 23.4% of these cases were not tested for CD4, only 14/206 (7.6%) with tested CD4 had CD4 >200, thus about 92% of tested pts, had CD4 <200. Thus, the sentence in the abstract should be modified, and it should be also mentioned in the Results and Discussion sections..

-It is well known that cotrimoxazole protects against toxo in immunosuppressed PLWH. The fact that in this study it seems that no differences in cotrimoxazole use was found between cases and controls, suggests that adherence to prophylaxis might be an issue in this population. Authors should comment this aspect in the Discussion section.

-In the multivariate analysis, variables have been adjusted for CD4 count? It may be that PLWH being IDU or HBV+ had lower CD4 count (as probably occurred with pts not receiving ART) and if adjusting results may vary.

-I would say immune reconstitution instead of immune reconstruction

6. PLOS authors have the option to publish the peer review history of their article (what does this mean?). If published, this will include your full peer review and any attached files.

Reviewer #1: **Yes: **Tun-Chieh Chen

Reviewer #2: No

---

## [Author Response · Author response to Decision Letter 0]

18 Dec 2023

Response to Reviewers’ comments

Reviewer #1

1. Because patient geographical distribution and infection period difference are also critical for toxoplasmosis infection, the manuscript needs to analyze these data.

Author’s response #1

We agree with the reviewer that geographical distribution and infection period differences are important in toxoplasmosis infection. Like the reviewer, we are aware of the importance of these two factors, and therefore have designed our study to analyse the site and time effect by matching by site and time period of toxoplasmosis infection so that cases and controls were compared within the same site and in a similar time frame. Matching by site and time period removed the known confounding effects of the two risk factors and allowed for comparison of cases and controls without the influence of these factors. Although matching does not output the effect sizes in the regression table for site and time period of infection (unlike the regular unmatched analyses), the regression model had “analysed” or taken into account these two factors by way of matching through the use of conditional logistic regression “conditioned on matched pairs”. Therefore, site and time were analysed by conditional logistic regression through matching. Unfortunately, we are not able to provide the number of cases by individual site to maintain confidentiality for the network, however we have listed the countries that were included, and have also further clarified the matching in our methods.

“To account for differences in site and time period, prospective cases were matched to prospective controls, while retrospective cases were matched to retrospective controls from the same site and time period.”

“Factors associated with toxoplasmosis were analyzed using conditional logistic regression to account for matching.”

“A total of 269/9576 (2.8%) PLWH were diagnosed with toxoplasmosis from a subset of 19 TAHOD sites from Cambodia, China, Hong Kong SAR, India, Indonesia, Japan, South Korea, Malaysia, the Philippines, Singapore Taiwan, Thailand, and Vietnam, and were matched to 538 controls.”

2. Because many patients have not taken ART and lack CD4 & viral load data, these data deficiencies could interfere with the risk factors assessment.

Author’s response #2

Thank you for your comment. As you said, it is disappointing that many of the subjects did not have CD4 or VL values. But while revising the paper this time, we excluded people without CD4 values, and found the vast majority (93.2%, instead of the previous 71%) were advanced HIV with a CD4 count of less than 200. Like your good opinion, most of the people who actually participated in the study lacked of data, but I think these results will be valuable.

3. The IVDU etiology and HBV infection should be considered as counfounding factors and these factors are highly associated to socioeconomic status which could be the risk factor of toxoplasmosis infection.

Author’s response #3

We would like to thank the reviewer for this suggestion. Injecting drug users (IDU) are normally known to be associated with hepatitis C co-infection, rather than hepatitis B. From our analysis, hepatitis C was not significant in the multivariate analysis. We have also tested for collinearity between mode of HIV exposure and HBV and found that there was no collinearity between the two variables (mean VIF=1.00). 

4. The disease type of toxoplasmosis should be described in this manuscript.

Author’s response #4

We have updated the Materials and Methods section as follows; (page 5, line 133)

Definition of toxoplasmosis 

We defined meaningful toxoplasma infection as definitive and presumptive toxoplasmosis, and definitive toxoplasmosis means patients who confirmed toxoplasmosis by microscopy (histology or cytology). Presumptive toxoplasmosis means patients whom suspicious toxoplasmosis of brain satisfying all three of the following. (i) Recent onset of a focal neurological abnormality consistent with intracranial disease or a reduced level of consciousness; (ii) Evidence by brain imaging (computed tomography or nuclear magnetic resonance) of a lesion having a mass effect or the radiographical appearance of which is enhanced by injection of contrast medium or consistent symptomatic presentation; (iii) Serum antibody to toxoplasmosis or successful response to therapy for toxoplasmosis.

 

Reviewer #2

1. Despite in the Methods section they comment that “controls without VL or CD4 unavailable within the matched timeframe were not selected because they were considered key covariates”, 62.5% and 83.8% of controls were not tested for CD4 or VL respectively (compared with 23.4% and 70.3% in cases). Could the authors clarify this relevant aspect of their study?

Author’s response #1

We apologize if this was unclear. We excluded controls without both CD4 AND VL, but allowed controls to be included if they had either a CD4 or a VL available, in order to maximize our sample size. We have clarified this in our methods:

“The controls were randomly selected and were required to have at least a CD4 or VL available within the matched timeframe as they were considered key covariates in the study. Controls without both viral load and CD4 available were not selected.”

2. In the abstract they say that 71% of toxo pts had CD4 < 200 , that means that 29% had >200, what is surprising as almost all PLWH with toxo have CD4 < 200 and mostly <100. But in fact, as 23.4% of these cases were not tested for CD4, only 14/206 (7.6%) with tested CD4 had CD4 >200, thus about 92% of tested pts, had CD4 <200. Thus, the sentence in the abstract should be modified, and it should be also mentioned in the Results and Discussion sections.

Author’s response #2

We would like to thank the reviewer for this suggestion. As you mentioned, I agreed to calculate the proportion of people whose CD4 values were measured, so we revised the abstract and result sections of the main text as follows, and additionally mentioned this in the discussion, as follows; (page 12, line 223)

“Excluding 63 out of 269 people without CD4 values, 192 (93.2%) had CD4 ≤200 cells/µL and 162 (78.6%) had CD4 ≤100 cells/µL.”

“Furthermore, CD4 value was lower in the toxoplasmosis group compared to the control. Excluding cases without CD4 test results, the majority had advanced HIV, with 93.2% having CD4 ≤200 cells/µL. Assuming the same seroprevalence as site matching was performed, low CD4 count is a prominent risk factor for overt toxoplasma infection. Therefore, immune reconstitution through early diagnosis and use of ART remains an important task.”

3. It is well known that cotrimoxazole protects against toxo in immunosuppressed PLWH. The fact that in this study it seems that no differences in cotrimoxazole use was found between cases and controls, suggests that adherence to prophylaxis might be an issue in this population. Authors should comment this aspect in the Discussion section.

Author’s response #3

Thank you for your comments. In fact, although the CD4 count was lower in the case group compared to the control, it was not significantly higher in those who took cotrimoxazole for prophylactic purposes. In fact, the overall cotrimoxazole administration before and after diagnosis was high in the case group. But there was no difference when analyzed with preventive purposes and it means that there may have been a problem with adherence, and this is an important result, so details about this have been added in the discussion section as follows; (page 12, line 243)

“Also, prophylactic cotrimoxazole administration did not contribute to lowering the risk of toxoplasmosis. When we first analyzed cotrimoxazole administration before and after the diagnosis of toxoplasmosis, cotrimoxazole administration was significantly high in the case group. But when limited to prophylactic use, there was no difference from the control group. This suggests that there might be a problem with medication compliance, but our study cannot prove it.”

4. In the multivariate analysis, variables have been adjusted for CD4 count? It may be that PLWH being IDU or HBV+ had lower CD4 count (as probably occurred with pts not receiving ART) and if adjusting results may vary.

Author’s response #4

The multivariate results were adjusted for CD4. 

5. I would say immune reconstitution instead of immune reconstruction

Author’s response #5

As suggested, we have modified the pointed words in Conclusion section.

---

## [Decision Letter · Decision Letter 1]

2 May 2024

PONE-D-23-29849R1Risk factors for toxoplasmosis in people living with HIV in the Asia-Pacific regionPLOS ONE

Dear Dr. Choi,

Thank you for submitting your manuscript to PLOS ONE. After careful consideration, we feel that it has merit but does not fully meet PLOS ONE’s publication criteria as it currently stands. Therefore, we invite you to submit a revised version of the manuscript that addresses the points raised during the review process.

We look forward to receiving your revised manuscript.

Kind regards,

Masoud Foroutan, Ph.D.; Assistant Professor

Academic Editor

PLOS ONE

Journal Requirements:

Reviewers' comments:

Reviewer's Responses to Questions

**Comments to the Author**

1. If the authors have adequately addressed your comments raised in a previous round of review and you feel that this manuscript is now acceptable for publication, you may indicate that here to bypass the “Comments to the Author” section, enter your conflict of interest statement in the “Confidential to Editor” section, and submit your "Accept" recommendation.

Reviewer #1: All comments have been addressed

Reviewer #3: (No Response)

2. Is the manuscript technically sound, and do the data support the conclusions?

Reviewer #1: Yes

Reviewer #3: (No Response)

3. Has the statistical analysis been performed appropriately and rigorously? 

Reviewer #1: Yes

Reviewer #3: Yes

4. Have the authors made all data underlying the findings in their manuscript fully available?

Reviewer #1: Yes

Reviewer #3: Yes

5. Is the manuscript presented in an intelligible fashion and written in standard English?

Reviewer #1: Yes

Reviewer #3: Yes

6. Review Comments to the Author

Reviewer #1: 1. Thanks for your responses to the questions, and then it makes the information from the manuscript more clear. However, the case distribution and diagnosis time period are also interesting for the readers. Although these factors were matched for the analysis. I recommend that you can provide information about the case's geographical and time period distribution in a small table or in the supplement material. It is much better for the reference of our readers.

Reviewer #3: Dear Author, please:

Use keywords based on MESH.

Mention the prevalence and risk factors in the title.

Many sources are old. Please use new sources from the last 5 years

7. PLOS authors have the option to publish the peer review history of their article (what does this mean?). If published, this will include your full peer review and any attached files.

Reviewer #1: **Yes: **Tun-Chieh Chen

Reviewer #3: No

---

## [Author Response · Author response to Decision Letter 1]

10 Jun 2024

Reviewer #1

1. Thanks for your responses to the questions, and then it makes the information from the manuscript more clear. However, the case distribution and diagnosis time period are also interesting for the readers. Although these factors were matched for the analysis. I recommend that you can provide information about the case's geographical and time period distribution in a small table or in the supplement material. It is much better for the reference of our readers.

Author’s response #1

We added the information about [Year of toxoplasmosis diagnosis] as supplementary 1 to the first paragraph of the “Demographics and clinical characteristics of the study population” of Result section.

Reviewer #3

1. Dear Author, please: Use keywords based on MESH. Mention the prevalence and risk factors in the title. Many sources are old. Please use new sources from the last 5 years

Author’s response #1

We modified the keywords to be MESH-based, and since prevalence cannot be inferred from our study, only risk factors are mentioned in the title. Lastly, we made some revisions referenced in this study.

---

## [Decision Letter · Decision Letter 2]

14 Jun 2024

Risk factors for toxoplasmosis in people living with HIV in the Asia-Pacific region

PONE-D-23-29849R2

Dear Dr. Choi,

We’re pleased to inform you that your manuscript has been judged scientifically suitable for publication and will be formally accepted for publication once it meets all outstanding technical requirements.

Kind regards,

Masoud Foroutan, Ph.D; Assistant Professor

Academic Editor

PLOS ONE

Additional Editor Comments (optional):

Reviewers' comments:

Reviewer's Responses to Questions

**Comments to the Author**

1. If the authors have adequately addressed your comments raised in a previous round of review and you feel that this manuscript is now acceptable for publication, you may indicate that here to bypass the “Comments to the Author” section, enter your conflict of interest statement in the “Confidential to Editor” section, and submit your "Accept" recommendation.

Reviewer #3: All comments have been addressed

2. Is the manuscript technically sound, and do the data support the conclusions?

Reviewer #3: Yes

3. Has the statistical analysis been performed appropriately and rigorously? 

Reviewer #3: Yes

4. Have the authors made all data underlying the findings in their manuscript fully available?

Reviewer #3: Yes

5. Is the manuscript presented in an intelligible fashion and written in standard English?

Reviewer #3: Yes

6. Review Comments to the Author

Reviewer #3: (No Response)

7. PLOS authors have the option to publish the peer review history of their article (what does this mean?). If published, this will include your full peer review and any attached files.

Reviewer #3: No

---

## [Editor Report · Acceptance letter]

21 Jun 2024

PONE-D-23-29849R2 

PLOS ONE

Dear Dr. Choi, 

I'm pleased to inform you that your manuscript has been deemed suitable for publication in PLOS ONE. Congratulations! Your manuscript is now being handed over to our production team.

Kind regards, 

on behalf of

Dr. Masoud Foroutan 

Academic Editor

PLOS ONE